

# FLWA
## FoodLink - A Food Donation Web Application



**Autors**: (Khalid Hassan Muzaffar) · (Mert Nuri Kodzhaaslan) · (Ziad Ayman Oun)

**Supervisor:** Krystian Wojtkiewicz

### Abstract

Foodlink is a web-based platform that provides a direct link between surplus food from households, restaurants, companies and charities for further distribution to people in need. The project has developed an automated matching platform based on data analytics and uses an interactive map to strengthen the effective and responsive donation of food to the needy. The effects of this system are the increased efficiency in the food delivery process mentioned above, the reduction of food waste, and also the introduction of accountability for redistributed food. Foodlink not only helps ensure food security as it redistributes supplies more efficiently, but also builds partnerships and provides the NGOs the platform to manage food stocks effectively. This has fulfilled the promise of technology-based solutions against both hunger and food waste.

## 1 INTRODUCTION

These two issues are interconnected. Food wastage is a huge concern throughout the world, and Food Link aims to address the problem of surplus food by going to waste when hunger exists and famine takes over millions. Food-link helps address this challenge by providing a user-friendly and effective Web-based service that enables food donors and recipients, including charities and nonprofit organizations, to find each other quickly and efficiently. The foundation of this problem lies in the disconnect between the amounts of surplus food and the existing structural and management capabilities to effectively redeploy surplus food.

Food-link's aim in the business and technical domain is to provide the modern world with the opportunity to create an interactive approach to the problem of effective redistribution of surplus food in order to reduce waste and improve food security. The importance of the current project is determined by its ability to integrate automated features and overcome frictions in the operational processes of an organization and to improve clarity so that organizations can easily engage in social impact projects.

The set objectives for the team included:

- The development of a web platform that will allow donors and recipients to connect based on automated matching.

- To provide NGOs and food businesses with the means that would make donations easier, more effective, and transparent.

- Development of a data analytics feature to monitor donation trends in order to effectively improve the allocation of resources.

- These range from effective usage of surplus food for commercial purposes, reduced the cost of wastage by the business entities, and increased public relations through participation in philanthropic activities. From the technical perspective of the project, the objective was to ensure that the solution would be solid and user-friendly, the operations would run smoothly, and there would be clarity of communication between different stakeholders.

## 2   RESULTS

The project successfully designed a system that would integrate the following key functionalities:

- **Automated Matching:** It will automatically link donors and recipients depending on the amount and type of food. This enables the surplus of food to be channeled in a structured way.

- **Claim Feature:** Allows NGOs to view and claim food donations directly. Through this feature, surplus food will reach beneficiaries with increased efficiency.

- **Data Analytics Dashboard:** Provides real-time insights into food waste trends and donation efficiency, indicating the overall success of the project.

- **Interactive Map:** It aids the donor and recipient coordinate with each other for smooth pickup and delivery. Therefore, donors are able to plan logistics in the best way to avoid delays.

- **Confirmation and Notification System:** Provides automatic notifications to both donors and recipients, easing communication and ensuring that both parties are informed about the status of donations. After claiming the food, either the vendor or the non-profit organization can confirm the transaction, providing further accountability and transparency.

The integrated functions in Food-link helped not only smooth the entire donation process but also make the information chain quite transparent for all parties involved. The real-time analytics and an interactive interface of the system lead to a reduction in food wastage and increased effectiveness of charity.

## 3   ACKNOWLEDGMENTS

We would like to thank our supervisor, Krystian Wojtkiewicz, for his invaluable guidance and support throughout the project.

## 4   TECHNOLOGIES USED

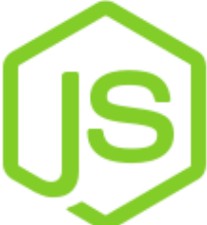 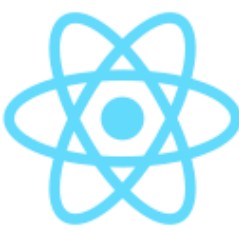 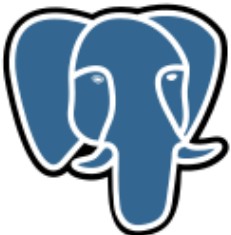

Figure 1: Technologies Used: Node.js, React, PostgreSQL

# 5   CLASS DIAGRAM

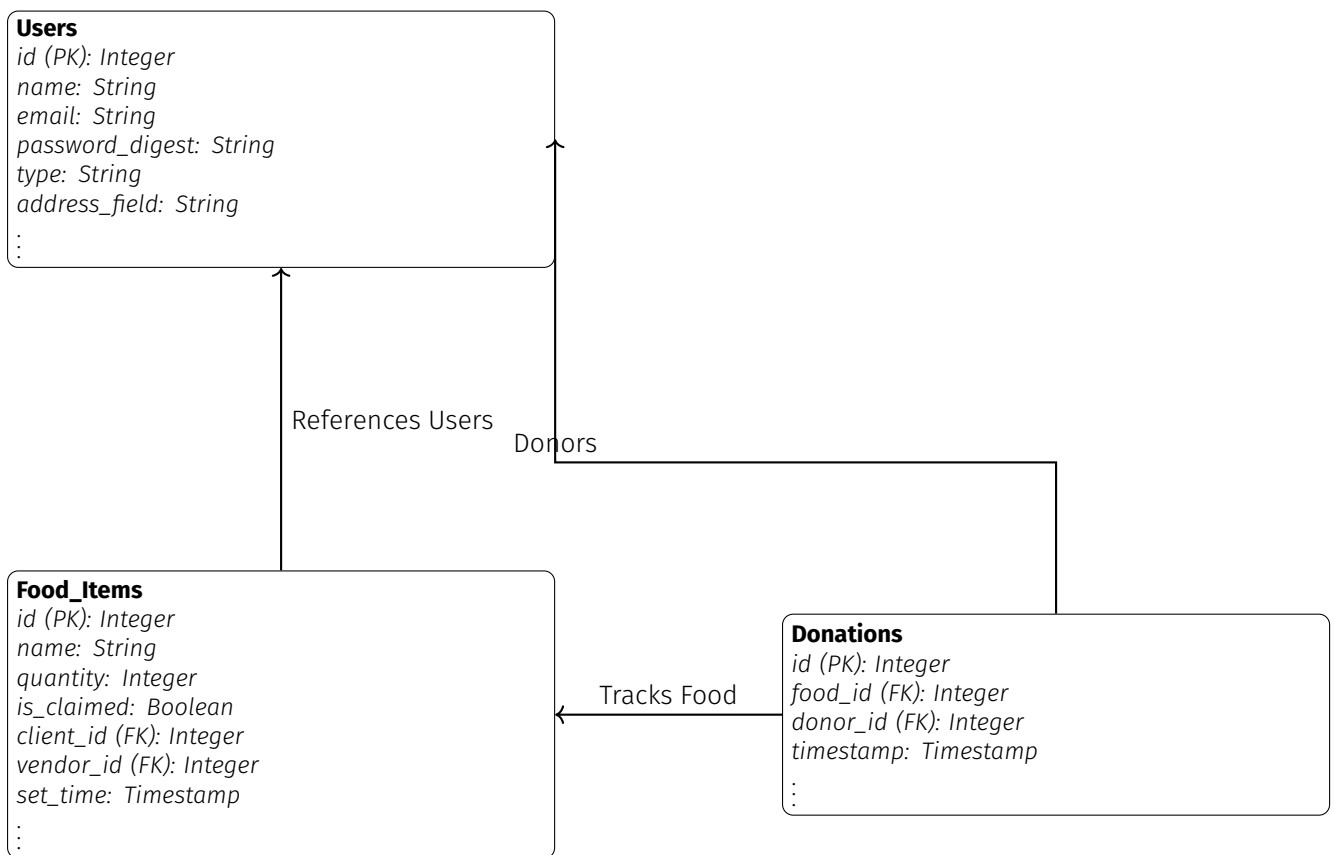

Figure 2: Class Diagram for Database Tables

# 6   SEQUENCE DIAGRAM

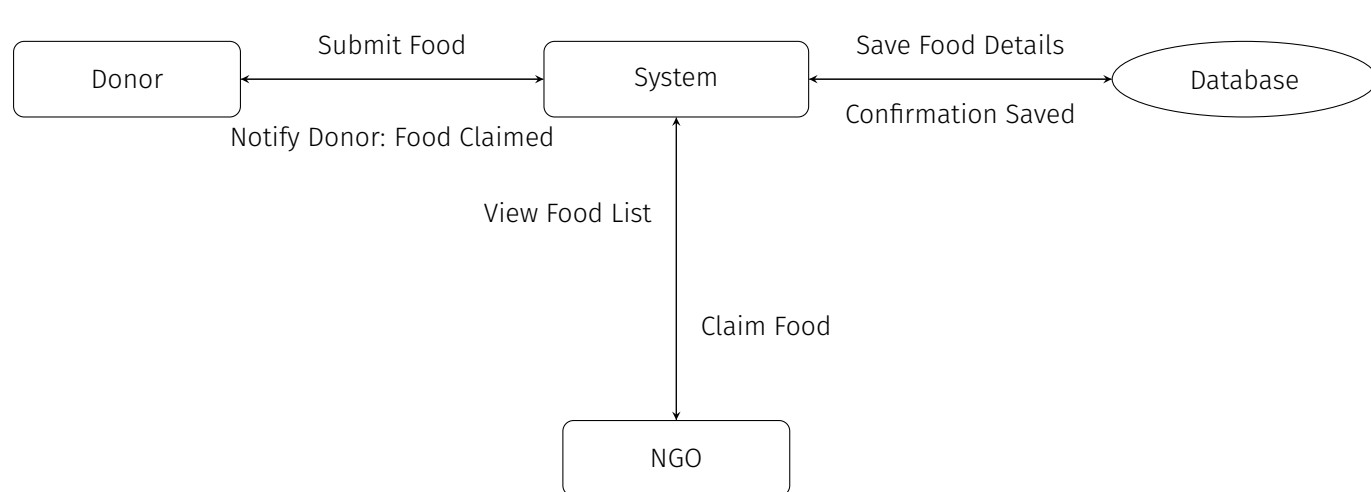

Figure 3: Improved Sequence Diagram for Donor, System, and NGO Workflow

## 7 CONTROL FLOW DIAGRAM

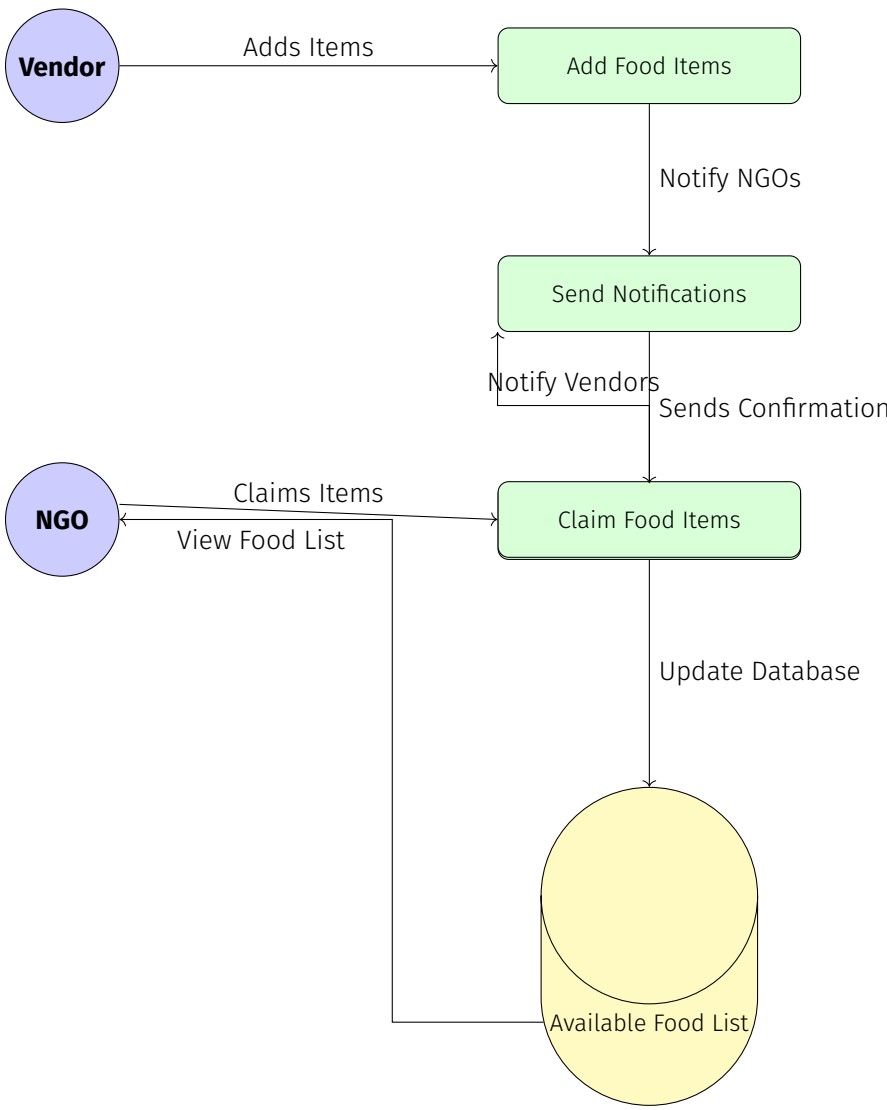

Figure 4: Enhanced Control Flow Diagram for Vendor and NGO Actions

## 8 CONCLUSIONS

Food-link has been highly promising in its potential to overcome the problem of hunger from food waste through technology-driven solutions. The result of this project provided a broadly beneficial platform for both donors and recipients alike, increasing efficiency for both food donations and waste reduction. For businesses, Foodlink has offered an avenue through which surplus could be managed effectively, while NGOs would have been assured of timely and proper targeting in distribution of this scarce resource of food. Probably the most important accomplishment of this project has involved the introduction of an automated, data-driven platform for optimized surplus food redistribution. The system offers effective matchmaking between donors and recipients, as well as real-time data that will help ensure transparency and precision to drive a legitimate impact on vulnerable communities' food security. Future Works: This would further involve, possibly, expanding the integration of the system with more local food businesses to increase the scope and efficiency of surplus food distribution. Demand forecasting using artificial intelligence would further tune the system for effective prediction and matching of food supply to areas with high demand. In addition to this, a mobile application version will increase accessibility of the platform by allowing the user to easily access the system while on the move. Similarly, partnerships with coordinating providers could allow delivery to be smoothed further.

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

Node.js is a powerful platform for building scalable applications [3]. Express provides a lightweight framework for web servers [2]. React is a popular library for building user interfaces [4]. Understanding food waste statistics is critical to combating global hunger [1].