# OpenReview forum: "FoodLink - A Food Donation Web Application"
_pwr.edu.pl/Wrocław_University_of_Science_and_Technology/2024/ZPI_Day — Wrocław University of Science and Technology 2024 ZPI Day Submission_

### Official Review · Reviewer_rbUU · 2024-12-03
**FoodLink - A Food Donation Web Application**

**Confidence:** 2
**Significance Of Results:** 3
**Overall Quality:** 3

**Compliance With Template:**

3: Average Quality – The article includes most of the required sections, but some may be incomplete, written in a general or unclear manner. The content is correct but requires further refinement.

**Description Of Results:**

3: Average Quality – The results are described with moderate detail. Some examples or evaluation elements are present but insufficiently developed or incomplete.

**Feedback On Consistency:**

Consistency of the project description i only adequate. Important parts like related works/solutions/projects is missing. Its not quite clear what is contribution and result of project.

**Potential For Development:**

Project for sure has potential for further work and practical application of its result.

**Project Nature Evaluation:**

Project exhibit characteristics of an engineering work, as it present implementing of simple web application. But level of utility, application of technical methods, and technological solutions is hard to assess

**Technical Language Precision:**

3: Average Quality – The language is mostly appropriate but may contain minor terminological or stylistic errors. Some statements might lack precision or require improvement for better readability.

---

### Official Review · Reviewer_6C7k · 2024-12-04
**Article of average quality - requires significant revisions before publication**

**Confidence:** 4
**Significance Of Results:** 3
**Overall Quality:** 3

**Compliance With Template:**

3: Average Quality – The article includes most of the required sections, but some may be incomplete, written in a general or unclear manner. The content is correct but requires further refinement.

**Description Of Results:**

3: Average Quality – The results are described with moderate detail. Some examples or evaluation elements are present but insufficiently developed or incomplete.

**Feedback On Consistency:**

The text in some places is chaotic and incoherent.
1. What does the FLWA in the title mean?
2. Why are the names of the authors in backets?
3. Strange wording: "The project has developed an automated"
4. Lack of Related work section.
5. Diagrams are lacking graphically (straighten the lines, review the titles: what does improved and enhanced mean)
6. Inconsistent wording: is vendor the same as donor? Is it FLWA, FoodLink, Food-link or Food Link?
7. What purpose do the icons on Figure 1 have? Would it be more beneficial to show some screenshots instead?
8. Summery of technologies after references seems quite strange at this location.

**Potential For Development:**

Yes it does. The opportunities for project development are well described (although somewhat hidden in the conclusions section).

**Project Nature Evaluation:**

The authors mention "automated matching platform" in the abstract. But the text lacks any details on how this functionality was implemented. Such a module, if developed right, would present a significant level of utility and could be a nice piece of engineering work done for a good cause. This needs to be further refined and described, because it seems like the biggest contribution of the project.

I do not understand the Control Flow Diagram. Are the food items really added to the database after the notification and claim? This seems very strange. But I might miss something here. Maybe this diagram could be used to describe the how the platform implements the matching.

**Technical Language Precision:**

3: Average Quality – The language is mostly appropriate but may contain minor terminological or stylistic errors. Some statements might lack precision or require improvement for better readability.

---

### Official Review · Reviewer_yrvx · 2024-12-05
**Great potential, minor results**

**Confidence:** 5
**Significance Of Results:** 3
**Overall Quality:** 2

**Compliance With Template:**

3: Average Quality – The article includes most of the required sections, but some may be incomplete, written in a general or unclear manner. The content is correct but requires further refinement.

**Description Of Results:**

2: Low Quality – The results are described very superficially and in a general manner. Essential details, usage examples, or evaluations are missing.

**Feedback On Consistency:**

Unfortunately, the description of the project is very minor. The introduction seems a bit generic. It does not indicate proper problem analyses or show that the authors underwent all project stages. Starting from the third section, authors follow the rule that a picture tells thousands of words, but they chose the extreme aspect of that.

**Potential For Development:**

Huge, as authors did not really focus on showing any real achievement.

**Project Nature Evaluation:**

It is hard to assess; it seems that there is/will be a web application. However, it is not clear if the application was actually developed.

**Technical Language Precision:**

1: Very Low Quality – The language is inappropriate for a technical report. It contains numerous imprecise statements, terminology errors, and incorrect expressions, significantly hindering comprehension.

---

### Decision · Program_Chairs · 2024-12-10

Accept (Poster)